# WHEN CAN TRANSFORMERS COUNT TO N?

## ABSTRACT

Large language models based on the transformer architecture can solve highly complex tasks. But are there simple tasks that such models cannot solve? Here we focus on very simple counting tasks, that involve counting how many times tokens in the vocabulary appeared in a string. We show that if the dimension of the transformer state is linear in the context length, this task can be solved. However, the solution we propose does not scale beyond this limit, and we provide theoretical arguments for why it is likely impossible for a size-limited transformer to implement this task. Our empirical results demonstrate the same phase-transition in performance, as anticipated by the theoretical argument. Our results demonstrate the importance of understanding how transformers can solve simple tasks.

## 1 INTRODUCTION

Large language models (LLMs) have demonstrated striking performance on a wide array of tasks, from creative writing to solving complex math problems. Given these successes, a key question arises: what can these models do, and just as importantly what can they not do. There are multiple ways to address this question of expressiveness of LLMs. First, it can be studied empirically by probing LLMs and seeking relatively simple tasks that they cannot perform successfully. Indeed recent work has found several such tasks, including "needle in haystack" (Kamradt, 2024; Ivgi et al., 2023) as well as extrapolating tasks to longer sequences (Levy et al., 2024). A second, complementary, approach is theoretical studies which chart the computational capabilities of transformers (Sanford et al., 2023; Wei et al., 2022a).

In the current work, we focus on a simple task that transformers often struggle with, and analyze it theoretically. Specifically, we consider a very simple "Query Counting" task, defined as follows. The model is presented with a sequence of tokens, and is then asked how many times a given query token appears in the sequence. For example:

> *Consider the sequence a a b b a c c d a. How many times does the letter "a"*
> *appear in the sequence?*

Our interest in this problem is that it seems like a basic module needed for working with long input streams. Indeed, there is a long line of work on sketching and streaming, that studies similar tasks (Alon et al., 1996; Cohen, 2014). Furthermore, this task can be viewed as an extension of the "needle in haystack" problem which has recently garnered much attention, since it was shown that most LLMs struggle with it for long context. In the needle in haystack problem, the goal is to find a specific string in a long text. In our counting problem, the model is tasked with counting how many times a given string has appeared, which is a harder task than finding one appearance.

There is, however, a key difference between the needle in haystack problem and the counting problem we consider above. The needle in haystack problem is clearly solvable by transformers, regardless of the context length. This is because detecting a similar token and extracting it (or nearby tokens) is a simple task for single attention head (e.g., using induction heads Olsson et al., 2022). On the other hand, for the counting problem, as we argue here, it is much less clear that transformers can solve it for arbitrary context length. Note that when we write that transformers cannot perform a task, we are referring to transformers whose number of parameters is not dependent on context size.

Modern LLMs indeed struggle with counting tasks (e.g., see Section 6 and (Barbero et al., 2024)). Of course if these models can use code, the task becomes easy, but our focus is on understanding the

capabilities of the transformer architecture itself. Specifically, we ask whether transformers have an architectural limitation related to the counting task.

We next turn to ask when transformers can count. We show a construction that works as long as the transformer embedding size $d$ is greater than the vocabulary size $m$. Our construction uses one-hot embeddings, or more generally orthogonal embeddings, of the vocabulary, which allow the model to maintain a histogram of counts of tokens previously observed. When $d < m$ this orthogonal construction is no longer possible. A natural approach is to consider the "most orthogonal possible" embedding (a notion formalized in Welch bounds (Welch, 1974)), and try to use it in a similar scheme. However, we show that this does not allow implementing the histogram solution if the dimension is smaller than the vocabulary.

The above discussion suggests that in the $d < m$ regime,[1] the naive histogram approach does not seem to work. Our study of this regime reveals both a positive and a negative result. On the positive side, we show that there does exist a construction that allows counting, which can be done with a single transformer layer. On the negative side, we prove that this construction requires an MLP width that grows with context size, meaning it is not applicable to arbitrarily long contexts. Indeed, when training transformers on these counting tasks, we find that they fail in the $d < m$ regime.

We next study a somewhat more complex counting task, which we refer to as "Most Frequent Element". Here we present the model with a sequence of tokens, and ask for the count of the most frequent token. This is the same as taking the maximum of the histogram of the counts. Similarly to the Query Counting task, we show that in this case, a solution exists for $d > m$ based on an orthogonal construction. However, for $d < m$ we show, using a communication complexity argument, that no solution exists for a one layer transformer. Thus again we obtain a phase transition for counting at $d = m$.

Taken together, our results reveal an interesting landscape for simple counting task, where the $d = m$ threshold separates between transformers that can count and those that cannot.

Our results highlight the importance of studying basic counting problems, and their dependence on vocabulary size. They also point to limits of solving seemingly simple problems using transformers, and further emphasize the advantages of using code as a tool to sidestep these issues.

## 2 RELATED WORK

Since the introduction of transformer architectures (Vaswani et al., 2017) and the success of LLMs, there has been much work on evaluating their performance on various tasks (e.g., see Srivastava et al., 2023). In particular, much recent work has explored performance on long context tasks, the dependence of accuracy on context length (Levy et al., 2024), and the ability of models to extrapolate to lengths beyond those seen at training (Anil et al., 2022).

The fact that models often do not perform well on these evaluations has prompted works that try to pinpoint the inherent limitations of transformer models. One line of work is to use computational complexity lower bound approaches to show that transformers of a certain size and architecture cannot implement a given function of interest. For example, Sanford et al. (2023) show that certain functions cannot be implemented without the transformer size scaling with input size. A similar limitation is also shown in (Peng et al., 2024) for certain compositions of functions.

A related line of work is to relate transformers to known complexity classes. For example it has been shown that transformers can efficiently approximate Turing machines (Wei et al., 2022a), and that transformers with bounded persision can only solve problems in uniform $TC^0$ (Merrill and Sabharwal, 2023). Chain-of-Thought (Wei et al., 2022b) has also been analyzed from an expressiveness viewpoint, demonstrating it can substantially improve the expressive power of transformers (Feng et al., 2024).

Our focus here is not on the general capabilities of transformers but rather on a specific, seemingly simple problem, and on the ability of transformers to solve it.

---

[1]Some of our results involve regimes such $d < cm$ for a small constant $c$. We leave $c$ out for simplicity.

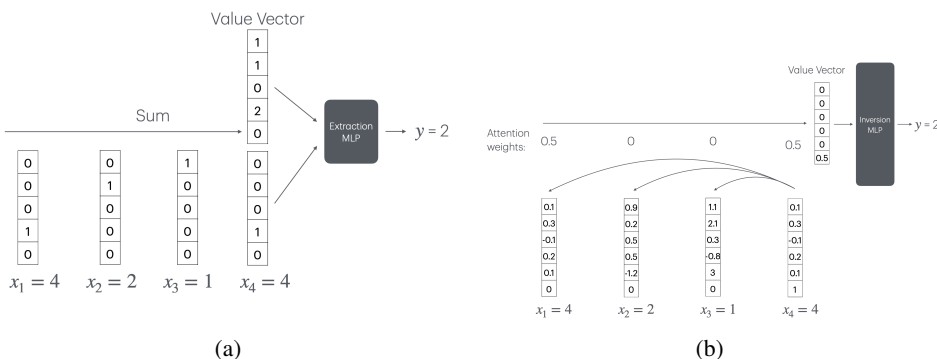

(a)                                             (b)

Figure 1: (a) Solving QC using a histogram (for $d > m$). To count the number of tokens with $x_i = 4$, we assume each token is embedded to the standard basis (this can be done because $d > m$, and sum these vectors across all input tokens. This results in a histogram of the inputs, and the $4^{th}$ element can be extracted using a simple "Extraction MLP".

(b) Solving QC using CountAttend: this solution works for all $d$, but requires an MLP for inverting numbers, and we show that this MLP need to be of size $n$ (which can be prohibitive). To count the number of tokens with $x_i = 4$, the last token attends to the others such that only tokens with $x_i = 4$ receive large weights. This results in weights that are non-zero only for $x_i = 4$, and the resulting weight on these is the inverse of the count of 4 (i.e., 0.5 in this case). Then this inverse is moved to the last element of the value vector, using a positional embedding coordinate that is 1 only for last token $n$. Finally, the inverse count needs to be inverted to get the desired count, and this requires the "Inversion MLP".

## 3 PROBLEM SETUP

We consider inputs that consist of a sequence of $n$ tokens: $x_1, \ldots, x_n$. The desired output for these is denoted by $y$. We consider the following two counting tasks: Query Count (QC) and Most Frequent Element (MFE) where $y$ is defined as follows:

- For the QC task: $y$ is the count of the token $x_n$ in the set $x_1, \ldots, x_n$ (i.e., $y \geq 1$ always).
- For the MFE task: $y$ is the count of the most frequent token in $x_1, \ldots, x_n$.

We denote the dictionary size by $m$, namely $x_i \in \{1, \ldots, m\}$. Furthermore, we use the following notations for model-related quantities:

- $d$: the key-size (i.e. embedding dimension of each head).
- $h$: the number of attention heads.
- $L$: the number of layers
- $p$: the numerical precision of computations: We assume that all arithmetic operations (including the softmax) are performed exactly using registers of $p$ bits.
- $D$: the overall embedding dimension, where $D = d \times h$
- The embedding of token $i$ is $\boldsymbol{v}_i \in \mathbb{R}^D$.
- The query, key, value matrices for layer $i$, attention head $j$ are denoted by $Q_{i,j}, K_{i,j}, V_{i,j}$. All are matrices in $\mathbb{R}^{dh \times d}$.
- $\boldsymbol{p}_i$: the positional embedding for location $i$.
- $\boldsymbol{u}_{i,j}$: the output of head $i$ in layer $j$.

Most of our solutions work with an architecture consisting of a single layer and a single head. When this is the case we omit the indices $i$ and $j$ from our notation. Also note that if $h = 1$ then $D = d$. In our theoretical results we do not use normalization layers, although we can easily add degenerate normalization that doesn't alter the input, as was done in (Sanford et al., 2023). In our empirical results we use standard transformer layers that do include normalization layers.

## 4 ANALYZING QUERY COUNT (QC)

In this section we focus on the QC problem, and ask which transformer architectures can implement it successfully. We first remark on a general limitation of transformers without positional embeddings for the counting problems we consider.[2] We next show in Section 4.2 that if $d > 2m$ a one-head one-layer transformer can implement QC. We refer to this as the histogram solution. We then show that the histogram solution stops working if $d < m$. The natural question is then whether there are other solutions for the $d < m$ case. We argue that in this case, solutions are likely to require calculating the function $1/x$, and show that this function would require an MLP layer of width $n$. This means we cannot expect the transformer to extrapolate to long context sizes, and therefore a one-layer transformer is unlikely to be able to implement QC, at least using the two most natural solutions.

### 4.1 THE NEED FOR POSITIONAL EMBEDDINGS

Transformers use self attention to average over previous representations. The fact that they average, rather than sum, leads to an interesting limitation on their ability to count. Specifically, it is easy to see that for variable context size, they cannot perform any counting task without the use of positional embedding. Consider the QC task and an input sequence $S_1 = x_1, \ldots, x_n$, where the goal is to return the count of $x_n$ in the sequence. Now consider the length $2n$ sequence $S_2 = x_1, \ldots, x_n, x_1, \ldots, x_n$. The correct output for this sequence is twice the correct output for $S_1$. However, a transformer without positional embeddings that is applied to $S_1$ will have exactly the same output as the one for $S_2$. This follows because averaging is invariant to input duplication.

The above restriction no longer holds when positional embeddings are used, and it is easy to see that it can be rectified with even a simple positional embedding that just signifies the last position (see our construction in Section 4.4). This implies that if a transformer has access to the legnth of the sequence, it may make it easier to count. Another thing to note is that while the above difficulty arises for counting, it does not arise if we are interested in calculating proportions (e.g., what is the fraction of the items of the sequence that are equal to $x_n$).

### 4.2 A "HISTOGRAM" SOLUTION FOR $d > 2m$

We begin by providing a solution for the case where the model dimension is larger than the vocabulary size.

**Theorem 4.1.** *For the Query Count problem and any context length $n > 0$, if $d > 2m$, there exists a transformer that solves it, which has one layer, one head, and an MLP with $d$ neurons.*

We provide the construction below (see also Figure 1a). We begin by describing it as a two head solution, but a one-head implementation is also simple using a residual connection. The idea is to construct a histogram of all previous tokens (i.e. the number of times each token appears) and then given the query token $x_n$ extract the count for this particular token from the histogram.

First, we assume that the embeddings are orthonormal. Namely:

$$\boldsymbol{v}_i \cdot \boldsymbol{v}_j = \delta_{ij} \qquad \forall i, j \in \{1, \ldots, m\} \tag{1}$$

where $\boldsymbol{v}_i$ is the embedding into $\mathbb{R}^d$ of the dictionary item $i$. This is possible because of the assumption $d > m$. For simplicity, we assume that $\boldsymbol{v}_i = \boldsymbol{e}_i$ where $\boldsymbol{e}_i$ is the standard basis in $\mathbb{R}^d$.

Next, we construct an attention head whose output at position $n$ is the histogram of the tokens up to and including this token. Let $Q_1 = 0$ (the zero matrix) and $V_1 = I_d$ ($I_d$ is the identity matrix in $\mathbb{R}^d$). Then the output of this attention head is

$$\boldsymbol{u}_1 = \sum_{i=1}^{n} \boldsymbol{e}_{x_i} = \sum_{j=1}^{m} c_j \boldsymbol{e}_j \tag{2}$$

where $c_j$ is the number of occurrences of item $j$ in the context normalized by $n$, the length of the context. That is $c_j = |\{i \in [n] \mid x_i = j\}|/n$. In words, $\boldsymbol{u}_1$ is a vector in $\mathbb{R}^d$ whose $i^{th}$ entry is the number of times that token $i$ appeared in the input $x_1, \ldots, x_n$.

---

[2]We note that a similar observation was made in (Barbero et al., 2024), we provide it here for completeness.

The second head is set to simply copy the input embedding. This is done by setting $Q_2$ and $K_2$ such that $K_2^\top Q_2 = T I_d$ where $T$ is sufficiently large and $V = I_d$. After this we have:

$$\boldsymbol{u}_2 = \boldsymbol{e}_{x_n} \tag{3}$$

The outputs of the two heads consist of the histogram and a one-hot identifier for the query token. Recall that our desired output is the count $c_{x_n}$ of the query token. We can extract this count using an MLP with $d$ ReLU gates in the hidden layer. Gate $i$ computes ReLU of $n \cdot u_1[i] - B \cdot (1 - u_2[i])$ for some sufficiently large constant $B$. It is easy to see that the output of gate $i$ is $c_{x_n}$ if $x_n = i$ and 0 otherwise.

**Remarks:** 1) Note that the above solution will only work for input of a fixed length $n$. As noted in 4.1, a transformer without positional embeddings (as the one in our construction) cannot possibly count inputs of variable lengths. Our construction here can be extended to variable lengths by using positional embeddings, together with an MLP for computing $1/x$. We elaborate on such an approach in the next section.

2) We can also implement the above histogram scheme with one head, by taking advantage of a residual connection. The idea is to use half of the coordinates in the embedding dimension to store the result of the attention module, and pass the original token to the MLP using the residual connection.

3) In the construction above, we assumed that $\boldsymbol{v}_i$ are the standard basis. However, a similar construction is possible when $\boldsymbol{v}_1, \ldots, \boldsymbol{v}_m$ are orthonormal, but not one-hot. In this case, the MLP will have to take the dot product of $n * \boldsymbol{u}_1$ and $\boldsymbol{u}_2$ to extract the count. This is less natural to do with RELU gates. One could, however, first apply to $\boldsymbol{u}_1$ and $\boldsymbol{u}_2$ a linear transformation (i.e., a rotation), that changes the basis to the standard basis and then extract the count as before.

### 4.3 THE HISTOGRAM SOLUTION BREAKS FOR $d < m$

Our solution in the previous section uses the fact that if $d > m$ we can embed the dictionary into orthogonal vectors in $\mathbb{R}^d$. When $d < m$ this is not possible. One may try to extend this solution by embedding the dictionary into a collection of "almost" orthogonal vectors. However any collection of $m$ $m \geq 2d$ vectors in $\mathbb{R}^d$ contains a pair of vectors whose inner product is at least $\Omega(1/\sqrt{d})$ in absolute value. This is a result of the Welch bounds (Welch, 1974) which provide upper bounds on the minimum dot product between $m$ unit-vectors in $\mathbb{R}^d$. This implies the following lower bound, which states that counting will fail in this regime.

**Theorem 4.2.** *Consider the "Histogram" solution for the counting problem presented in Section 4.2, and embedding vectors $\boldsymbol{v}_1, \ldots, \boldsymbol{v}_m$. For an input $\bar{\mathbf{x}} = (x_1, \ldots, x_n)$ to the counting problem, denote by $c_{x_n}$ the correct solution and by* `hist`$(\bar{\mathbf{x}})$ *the output of the "Histogram" solution.*[3] *If $m \geq 2d$, then for any embedding vectors $\boldsymbol{v}_i$'s there are inputs to the counting problem for which:* $|$`hist`$(\bar{\mathbf{x}}) - c_{x_n}| \geq 0.25\sqrt{n}$.

*Proof.* Let $\boldsymbol{v}_1, \ldots, \boldsymbol{v}_m \in \mathbb{R}^d$ with $m \geq 2d$, and let $A = \max_{i \neq j} |\langle \boldsymbol{v}_i \cdot \boldsymbol{v}_j \rangle|$. Assume without loss of generality that $A = \boldsymbol{v}_1 \cdot \boldsymbol{v}_2$. By the Welch bounds (Welch, 1974) for $k = 1$ we have that $A \geq \frac{1}{\sqrt{2d-1}}$. Consider the input $x_1, \ldots, x_n$ to the counting problem where $x_1, \ldots, x_{n-c}$ are equal to the same token which is different from $x_n$ and mapped to the embedding $\boldsymbol{v}_1$, and $x_{n-c}, \ldots, x_n$ are all equal to $x_n$ which is mapped to embedding $\boldsymbol{v}_2$. Then the output for the histogram solution is:

$$|\text{hist}(\bar{\mathbf{x}})| = |\langle (n-c)v_1 + cv_2, v_2 \rangle|$$

$$\geq c + \frac{n-c}{\sqrt{2d-1}} .$$

By choosing $c = 0.5n$ and $n = d$ we have the desired result. □

The theorem implies that even if the dictionary size is only linear in the embedding dimension, any solution will incur an error that grows with the context size. In practice, the dictionary can contain

---

[3]The Histogram solution in this case is $\boldsymbol{v}_{x_n} \cdot \sum_j c_j \boldsymbol{v}_j$, which is the natural generalization to the non-orthogonal case. As noted in Remark 3, it is generally more elaborate to implement because of the dot product, but as Theorem 4.2 shows, this solution has an inherent limitation, irrespective of this implementation difficulty.

millions of tokens, while the embedding dimension is at most a few thousands, thus this error can be quite large. Note that picking the embedding vectors at random (e.g. i.i.d Gaussians) will result in an even greater error than what is stated in the theorem, since the inner product between each two vectors will be larger (with high probability) than the lower bound we used in the proof.

## 4.4 THE COUNTATTEND SOLUTION FOR ALL $d$ AND $m$

In the previous section we considered a histogram based solution, which required $d > m$. Here we provide an alternative approach to solve the counting problem which works for any $d$ and $m$. However, as we shall see, this solution requires a large MLP, that must scale with the length of the input $n$. As a result, a transformer with a fixed MLP size will not be able to learn such a solution that will work with arbitrary $n$ values.

We first present a high level description of this construction, which uses a 1-layer transformer with a single head. The idea explicitly uses attention to count (hence the name CountAttend) as follows (see also Figure 1b). Assume that the query token is $x_n = 4$, so that we are seeking the number of elements in $x_1, \ldots, x_n$ that are equal to $4$, and assume that the number of these elements is $7$. Then the token $x_n$ can attend to all other tokens, such that attention weight is high for all $i$ such that $x_i = 4$ (and same for all these) and near-zero otherwise. Thus, the attention weight will be $\frac{1}{7}$ for all the $x_i = 4$ tokens, including $x_n$. We next need to extract this number and invert it, in order to get the answer $7$.

Extracting the attention weight can be done by using a single coordinate in the positional embedding that is one for position $n$ and zero otherwise. The value aggregation of self-attention can then extract the number $\frac{1}{7}$ in this coordinate. Finally, to invert this number we need an MLP to implement the function $1/x$. If the smallest number we need to invert is $1/n$ then this can be done with an MLP with $4n$ neurons.

The following proposition, proved in Appendix A, summarizes the properties of this construction.

**Proposition 4.3.** *For any $d, m, n$ there exists a transformer that solves the corresponding QC problem. The transformer has one layer, one attention head, dimension $d$, and an MLP of size $O(n)$. Furthermore, its matrix $K^\top Q$ is diagonal with elements of magnitude $O(\log n)$.*

### 4.4.1 LIMITATIONS OF THE COUNTATTEND SOLUTION

The advantage of Proposition 4.3 is that it works for any dimension $d$, and does not restrict $d$ to be larger than $m$ as in the histogram solution. However, the solution in Proposition 4.3 has two major limitations compared to the histogram solution presented in Section 4.2. We discuss these below.

First, Proposition 4.3 has an $O(n)$ sized MLP, which is a result of its internal implementation of the function $x \mapsto \frac{1}{x}$ using an MLP, where $x \in \left[\frac{1}{n}, 1\right]$, and the desired precision is $0.5$ (because we are counting, and can round the result). In the proof of 4.3 we used a naive implementation of this function. It is natural to ask if a smaller implementation exists. The following result shows this is not possible.

**Lemma 4.4.** *Any $2$-layer MLP with ReLU activations that approximates the function $f(x) = 1/x$ in the interval $x \in \left[\frac{1}{n}, 1\right]$ to within $L_\infty$ error of less than $1/2$ has $\Omega(n)$ neurons.*

*Proof of Lemma 4.4.* Let $g$ be a piecewise linear approximation of $f(x)$. Then for for $x = 1/k$, $k = 1, \ldots, n$, we must have $k - 1/2 \le g(x) \le k + 1/2$.

Consider the line $\ell(x_1, x_2)$ between $(1/x_1, x_1)$ and $(1/x_2, x_2)$ for some integers $x_1$ and $x_2$, $1 \le x_1, x_2, \le n$. The equation of this line is $y = (-x_1 x_2)x + x_1 + x_2$. Let $x_1 = k$ and $x_2 = k - c$ for some constant $c$ that we determine below. Then the equation of $\ell(k, k-c)$ is $y = -k(k-c)x + 2k - c$. Let $\ell'(k, k-c)$ be the line $y = -k(k-c)x + 2k - c - 0.5$ which is parallel and below $\ell(k, k-c)$. We claim that the point $A = (1/(k - c/2), k - (c/2) + 0.5)$ lies below $\ell'(k, k-c)$. By convexity this implies that $g$ must have a breakpoint between $1/k$ and $1/(k-c)$.

To prove the claim we have to show that

$$k - (c/2) + 0.5 \le -k(k-c)\frac{1}{k - (c/2)} + 2k - c - 0.5$$

It is easy to check that this holds for $c = 3$ and any $k$. This shows that $g$ must have at least $\Omega(n)$ linear pieces. Note that any 2-layer MLP with ReLU activations with $\ell$ neurons is a piecewise linear function with at most $2\ell$ pieces. This is because each ReLU neuron is a piecewise linear function with at most 2 pieces, and the MLP is just the sum of those neurons. □

Note that although the lemma focuses on 2-layer MLPs, it can be readily generalized to deep MLPs, e.g. using the lower bound on the number of linear pieces for deep ReLU networks from Telgarsky (2016). Although deeper networks can have more linear pieces using fewer neurons than shallow network, the depth would still need to scale with $\log(n)$ which is infeasible in practical implementations.

The second limitation of Proposition 4.3 is that the magnitude of its attention matrices scales logarithmically with the context size $n$. Since the temperature is inside the exponent, it means that the magnitude of the gradient should scale polynomially with the context size. This is possible given high-precision computational resources. However, transformers are trained with limited precision (e.g. 8- or 16-bit) which can make the optimization of such large weights infeasible.

Taken together the two observations in this section suggest that the despite the fact that QC has a transformer based implementation with one layer, this representation is too large to be applicable to arbirary $n$, and also potentially suffers from optimization difficulties.

## 5 ANALYZING MOST FREQUENT ELEMENT

In this section we consider the task of finding the most frequent element (MFE) in a given sequence of tokens. This problem is very much related to the counting problem, as intuitively it requires to count every token separately and compute the token that appears the maximum number of times. We show that there are strict bounds on the size of the embedding compared to the size of the dictionary, in order for transformers to be able to perform this task.

### 5.1 MFE MODELS MUST HAVE $d \geq \Omega(m)$

We begin by showing a lower bound on the required size of a 1-layer transformer solving the MFE task. The following result establishes that MFE can be implemented only when $dhp = \Omega(\min\{m, n\})$. This means that at a fixed precision $p$ and if $n > m$, the dimension $d$ must grow linearly in the vocabolary size in order to implement MFE. This is the same trend that we saw for the QC problem.

**Theorem 5.1.** *Suppose that there is a 1-layer transformer with $h$ heads, embedding dimension $d$, and $p$ bits of precision, followed by an MLP of arbitrary width and depth, that solves the MFE task for sequences of length $n$. Then, we must have that $dhp \geq \Omega(\min\{m, n\})$, where $m$ is the vocabulary size.*

The full proof can be found in Appendix B. The proof uses a communication complexity argument that is inspired by a lower bound of Sanford et al. (2023). This lower bound implies that transformers which solve the MFE task need to have an embedding size that scales with the size of the dictionary, or have many attention heads. Note that increasing the size of the MLP which follows the attention cannot break the lower bound.

### 5.2 MFE CAN BE IMPLEMENTED WHEN $d = O(m)$

The previous result showed that MFE cannot be implemented with a one layer transformer when $d$ is smaller than $m$. Here we show that it *is* possible to implement MFE when $d = O(m)$. This implies that $d = O(m)$ is tight for the MFE problem. The result is described below.

**Theorem 5.2.** *There exists a 1-layer transformer that solves the MFE task for sequences of size $n$ and dictionary size $m$, where the parameters $d, h, p$ are equal to: $d = m$, $h = 1$, $p = \log(n)$, and the MLP has $d^2$ neurons.*

The construction is again based on the histogram approach. Because $d > m$, one can compute the histogram of counts in the last position (as for QC). The only part left to be done is to extract the

maximum from the histogram, which can be done via a one layer MLP with $m^2$ units (each unit performs a maximum between two distinct elements in the histogram).

One limitation of the above result is that it requires an MLP that grows with $m$. This can be avoided if using two layers of attention. A two layer implementation is simple: use the first layer to calculate the "Query Count" for each element, and then use softmax to calculate the maximum over tokens. This construction does not need an MLP at all. Another option is to use a depth $\log(m)$ MLP with $m$ neurons at each layer to calculate the maximum from the histogram (see Safran et al., 2024), however having depth which relies even logarithmically on the dictionary size is infeasible for practical implementations.

To summarize the above results, we have shown that MFE cannot be implemented by a one layer transformer if $d < m$, and that if $d > m$ MFE can be implemented either with a one layer transformer with wide MLP, or a two layer transformer without an MLP.

## 6 EXPERIMENTS

Our analysis considers the dependence between the transformer model size $d$, and its ability to perform counting tasks. Specifically, we show that for vocabulary size $m$ that exceeds $d$, exact counting is likely to become impossible. In this section we perform experiments that support this observation. We begin with results for training a model from scratch and then also consider results with a pretrained LLM (Gemini 1.5).

### 6.1 TRAINING FROM SCRATCH

**Tasks:**   We consider the two counting tasks described in the text: Most Frequent Element (MFE) and Query Count (QC). We generate instances of these by sampling sequences of length $n$ uniformly from a set of $m$ tokens. Denote each such sequence by $x_1, \ldots, x_n$. The expected output $y$ for these is as follows:

- For the QC task: $y$ is the count of the token $x_n$ in the set $x_1, \ldots, x_n$ (i.e., $y \geq 1$ always).
- For the MFE task: $y$ is the count of the most frequent token in $x_1, \ldots, x_n$.

During training and evaluation we sample batches from the above distribution. Evaluation used 1600 examples in all cases.

**Model:**   We train a transformer model with the standard architectural components (self attention, MLP, layer norm, etc.). We use two layers and four heads (theoretically we could have used less, but optimization was faster with this architecture). Training uses Adam for optimization, with batch size 16 and step size $10^{-4}$. Training is run for $100K$ steps. Positional embeddings were optimized. For predicting the count $y$, we use a linear projection on top of the embedding of the last token in the last layer (i.e., we do this instead of the vocabulary prediction). Training was done via colab and takes about 15 minutes per model with the standard GPU provided therein.

**Parameter Settings:**   We experimented with several values of $d$ (between 8 and 128). For each, we varied $m$ in order to test dependence on vocabulary size (we use 20 values between $m = 5$ and $m = 150$). In order to keep the average count at a constant value of $c$, we set $n = cm$. We used $c = 10$ in all experiments.

**Results:**   Our focus is on understanding the dependence between $d$ and $m$ and the ability to count. Thus, we report results as follows. For each value of $d$, we find the value of $m$ at which counting begins to fails. Specifically, we consider $m$ at which counting accuracy falls below 80% . We refer to this as $m_{thr}(d)$. Figure 2a shows these for the two counting tasks. It can be seen that in both cases, the threshold indeed increases roughly linearly with $d$, agreeing with our theoretical analysis.

### 6.2 EVALUATION OF A PRETRAINED LLM

Our theoretical results highlight the role of vocabulary size in counting problems. Here we provide an exploration of this role in a trained LLM, Gemini 1.5. We provide the model with the query

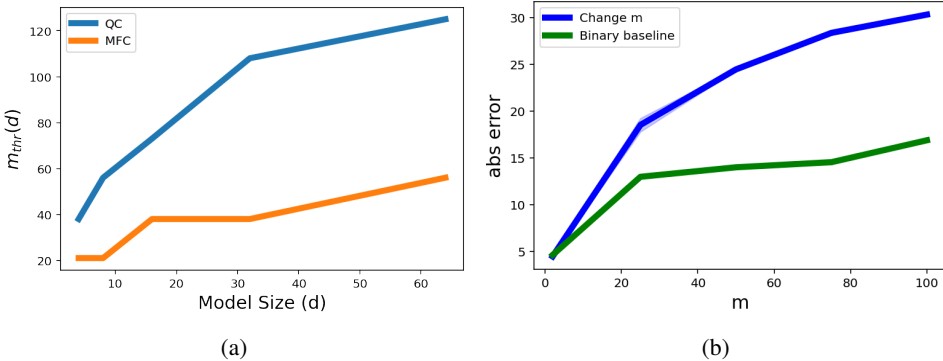

(a)                                                        (b)

Figure 2: (a) The threshold vocabulary size at which counting accuracy drops below $80\%$. Results shown for two counting tasks. (b) Results for the QC task when using Gemini 1.5. The x axis is the vocabulary size (i.e., the number of different tokens used in each sequence), the y axis is average absolute error over $500$ repetitions (standard error shown on curve). The "Binary Baseline" curve shows results when using just two tokens, but at the same sequence length used for the "Variable Vocab Size" curve. Standard errors also shown in shade.

count task.[4] We then vary $m$, the number of different words used in the sequences (e.g., for $m = 5$ we'll use a sequence with just five unique words), while keeping the expected counts of all elements at a constant $c = 40$. Namely, for each $m$ we use context length $mc$. The set of $m$ unique words are $m$ numbers sampled without replacement from $\{1, \ldots, 1000\}$. As a baseline for these, we also use the same sequence length, but with binary sequences matched to have expected count $c$ for the query token.[5] This allows us to estimate the error attributable to just the vocabulary size and not the sequence length and count. Results are shown in Figure 2b and it can be seen that increasing vocabulary size indeed has a negative effect on performance. Furthermore, this effect cannot be explained just by increasing the sequence size, since the binary curve is lower. Additional results are provided in the Appendix.

## 7 CONCLUSION

We focus on the basic task of counting using a transformer architecture. When the dimension of the model is sufficiently large, we show this task can be easily implemented by letting the transformer calculate the histogram of the input sequence. For smaller dimensions, we provide theoretical support suggesting that a one layer transformer cannot implement this function. Our empirical results support this phase transition.

Understanding such limitations of transformers are key to developing new architectures. For example, our results show that in a sense it would be impossible to have transformers count arbitrarily well and for long contexts, without increasing the architecture size considerably. Concretely, this suggests that for counting tasks it may be important to delegate to tools (Schick et al., 2024) such as code execution that do not have the same limitations.

## 8 LIMITATIONS

While we provide the first results on upper and lower bounds for counting, these are not yet tight, which would have been the ideal result. Specifically, we show impossibility for $d < m$ for MFE with one layer, but do not show that with more layers (e.g., two), though we conjecture this is true. Proving it would require novel technical tools, as it is not clear that the communication complexity argument is extendible to this case. For QC, we show that the inversion based architecture has inherent limitations for one layer, and here too it would be interesting to prove lower bounds for

---

[4]Specifically we use the prompt: "consider the following array [1,1,2,2,3] of length 5. How many times does the word 3 appear in the array? Respond in just one number. No additional text.".

[5]The two words used for these sequence are also sampled from $\{1, \ldots, 1000\}$.

additional layers. In terms of empirical evaluation, we restricted our training experiments to small architectures, but it would be interesting to explore these tasks for models closer to those used in practice. Additionally, it would be interesting to see how pretrained models perform on these tasks after fine-tuning on the task. Finally, it would be interesting to use a mechanistic interpretability approach to check which computation is actually being implemented in trained transformers (either pretrained on language or from scratch on counting).

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

## A  PROOFS FROM SECTION 4.4

*Proof of Proposition 4.3.* Here we provide additional information about the CountAttend solution. Recall that the idea is for the last token $x_n$ to attend to earlier tokens such that tokens identical to $x_n$ will eventually receive a weight close to $1/c_{x_n}$, the count of $x_n$ in the sequence. In what follows, we consider the scale of the logits that will provide this result at sufficient precision.

The attention weight of a unit-norm token embedding $\boldsymbol{v}_i$ with itself is $e^{T\boldsymbol{v}_i \cdot \boldsymbol{v}_i} = e^T$, and the attention weight of $v_i$ with $v_j$ is $e^{T\boldsymbol{v}_i \cdot \boldsymbol{v}_j} \le e^{TJ}$ where $J$ is an upper bound on the dot product between any two vectors in $\mathbb{R}^d$ among a set of $m$ vectors (e.g., as obtained from analysis of random vectors as in Section C).

Now consider the sum of the attention weights (i.e. the denominator of the softmax in the attention module). Let $c_{x_n}$ denote the number of occurrences of $x_n$ in the context, and let $c'$ denote the number of tokens $x_i$ such that $x_i \ne x_n$. We get that the sum of the attention weights is $c_{x_n} e^T$ plus a quantity bounded by $c' e^{TJ}$. If we divide this by $e^T$ then we get that the normalization factor equals to $c_{x_n}$ plus "noise" bounded by $c' e^{T(J-1)}$. From this we can recover $n_0$ if $c' e^{T(J-1)} < \frac{1}{2}$. We clearly satisfy this inequality if

$$T \ge \frac{\log(2n)}{1-J}.$$

Substituting the bound we have for $J$ for a random embedding (see Section C) we get that we need $T$ such that:

$$T = \Omega \left( \frac{\log(2n)}{1 - \sqrt{\frac{\log m}{d}}} \right).$$

Using the above, we obtain that the output of the attention is $1/c_{x_n}$ to within 0.5 accuracy in the inverse. To get $c_{x_n}$ we need an MLP that inverts $1/x$. This can be done as follows.

It is well known that we can implement a "delta function" using four ReLU neurons. For example we can approximate a delta function of height $h$ between $a$ and $b$, by $\frac{h}{\epsilon}(\max(0, x-a) - \max(0, x - (a + \epsilon)) - \max(0, x - b) + \max(0, x - (b + \epsilon)))$ for some sufficiently small $\epsilon$. We use 4 ReLU neurons to implement a "delta function" between $1/(k - 1/2)$ and $1/(k + 1/2)$ of height $k$ for each $k = 1, \ldots, n$. □

## B  PROOF FROM SECTION 5

*Proof of Thm. 5.1.* Our proof relies on the following set disjointness lower bound (Yao, 1979). (It is similar to a lower bound argument in Sanford et al. (2023), but simpler since we assume that all arithmetic in the transformer is performed exactly by registers of $p$ bits.) Alice and Bob are given inputs $a, b \in \{0, 1\}^n$, respectively. Their goal is to compute $\max_i a_i b_i$ by sending single bit

messages to each other in a sequence of communication rounds. The lower bound says that any deterministic protocol for computing $\max_i a_i b_i$ must have at least $n$ rounds of communication.

We construct a reduction from the set disjointness problem to the MFE task. We assume for ease of notation that the length of the context is $2n$, and also assume that $m > 3n$. If $m < 3n$ then we set the context size to be $n' = m/6$ and continue the proof as is with $n'$ instead of $n$. Note that since the lower bound is given by $\Omega(\min\{m, n\})$, using $n'$ instead of $n$ will provide a lower bound that depends on $m$. In fact, $\min\{m, n\}$ can be viewed as the "effective" dictionary size, which is the maximal number of different tokens that a transformer sees given an input sequence of length $n$.

Assume that Alice and Bob received inputs $a, b \in \{0, 1\}^n$. Suppose we have the following distinct tokens in our dictionary (which is possible by our assumption on $m$): $s_1, \ldots, s_n, y_1, \ldots, y_n, z_1, \ldots, z_n$. We consider the following input context $x_1, \ldots, x_{2n}$ to the transformer. For $j \in \{1, \ldots, n\}$, if $a_j = 1$ we set $x_j = s_j$, and otherwise we set $x_j = y_j$. Similarly, for every bit in $b$ in place $j \in \{1, \ldots, n\}$, if $b_j = 1$ we set $x_{n+j} = s_j$, otherwise we set $x_{n+j} = z_j$. We also assume there is some query token $x_0$ known to both Alice and Bob and different from the rest of the tokens. This is the last token and we assume that the desired output token corresponds to it.

Note that if the most frequent element in the context $x_1, \ldots, x_{2n}$ appears twice, then this token must be $s_\ell$ for some $\ell \in \{1, \ldots, n\}$ which means that $a_\ell b_\ell = 1$. Otherwise if the most frequent element appears only once and then $\max_i a_i b_i = 0$.

Suppose there exists a 1-layer transformer with $h$ heads followed by an MLP of arbitrary size that solves the MFE task for all inputs $x_1, \ldots, x_{2n}$. Assume the embedding dimension of each token is $d$, namely $s_i, y_i, z_i \in \mathbb{R}^d$ for every $i \in \{1, \ldots, d\}$. Also, denote the weights of the heads by $Q_j, K_j, V_j$ for each $j \in [h]$, and assume w.l.o.g. that they are of full rank (i.e. rank $d$), otherwise our lower bound would include the rank of these matrices instead of the embedding dimension (which can only strengthen the lower bound). We design a communication protocol (following the construction in (Sanford et al., 2023)) for Alice and Bob to solve the set disjointness problem:

1. Given input sequences $a, b \in \{0, 1\}^n$ to Alice and Bob respectively, they calculate $x_1, \ldots, x_n$ and $x_{n+1}, \ldots, x_{2n}$, respectively.

2. Alice computes the $p$ bit representation of

$$s_{j,a} = \sum_{i=1}^n \exp(x_i^\top K_j^\top Q_j x_0) \, ,$$

for each head $j$ and transmits them to Bob. The number of transmitted bits is $O(ph)$.

3. Bob finishes the computation of the softmax normalization term for each head $j \in [h]$ and sends it to Alice, namely he computes:

$$s_j = s_{j,a} + \sum_{i=n+1}^{2n} \exp(x_i^\top K_j^\top Q_j x_0) \, .$$

The number of transmitted bits is again $O(ph)$.

4. For each head $j \in [h]$ Alice computes the first part of the attention matrix which depends on her input tokens and transmits it to Bob. Namely, she computes:

$$t_{j,a} = \frac{\sum_{i=1}^n \exp(x_i^\top K_j^\top Q_j x_0) V_j x_i}{s_j} \, .$$

The number of transmitted bits is $O(dph)$, since $x_i \in \mathbb{R}^d$, and the assumption that $V_j$ is full rank.

5. Bob can now finish the computation of the attention layer. Namely, he computes:

$$t_j = t_{j,a} + \frac{\sum_{i=n+1}^{2n} \exp(x_i^\top K_j^\top Q_j x_0) V_j x_i}{s_j} \, .$$

Finally, Bob passes the concatenation of the vectors $t_j$ for $j = 1, \ldots, h$ through the MLP. This step does not require any additional communication rounds.

By the equivalence between the set disjointness and the most frequent element problem that was described before, Bob returns 1 iff the inputs $\max_i a_i b_i = 1$, and 0 otherwise. The total number of bits transmitted in this protocol is $O(dph)$, hence by the lower bound on the communication complexity of set disjointness we must have that $dph \geq \Omega(n)$. $\qquad\square$

*Proof of Thm. 5.2.* For the case of $d = m$, we consider the embedding vectors to be equal to $e_i$ for each token $i \in [m]$, namely the standard unit vectors. We use a single attention head with query matrix $Q = 0$, and value matrix $V = I$. In this case, it is easy to see that for any input sequence $x_1, \ldots, x_n$, the output of the attention layer is a vector $\boldsymbol{v} \in \mathbb{R}^d$ which is the histogram over the different tokens. Namely, if token which is mapped to $e_i$ appeared $c_i$ times, then $(v)_i = c_i$.

To find the most frequent token, we only need an MLP that outputs the maximum over a vector of numbers. To do so we can use the construction from (Safran et al., 2024). Namely, a one hidden layer MLP with width $O(d^2)$ (Thm 3.3 therein).

$\qquad\square$

## C   Inner Product of Random Vectors

Let $\boldsymbol{v}_1, \ldots, \boldsymbol{v}_m$ be random unit vectors in $\mathbb{R}^d$ where each coordinate in any vector is $\pm 1/\sqrt{d}$ with probability $1/2$, independently. Hoeffding's inequality (C.1) implies that with probability $1 - 1/poly(m)$, $|\boldsymbol{v}_i \cdot \boldsymbol{v}_j| = O(\sqrt{\frac{\log m}{d}})$ for all pairs $i, j \in [m]$, $i \neq j$.

Indeed, $\boldsymbol{v}_i \cdot \boldsymbol{v}_j$ is a sum of $d$ random variables of values $\pm 1/\sqrt{d}$ so by (4) we have

$$Pr\left(\boldsymbol{v}_i \cdot \boldsymbol{v}_j \geq t\right) \leq 2e^{-\frac{dt^2}{2}}$$

and therefore for $t = O(\sqrt{\frac{\log m}{d}})$ we get that the dot product is large than $t$ with polynomialy small probability for all pairs $\boldsymbol{v}_i, \boldsymbol{v}_j$.

**Proposition C.1.** *Hoeffding's inequality states that if $X_1, \ldots, X_n$ are independent random variables such that $a_i \leq X_i \leq b_i$ then*

$$Pr\left(|S_n - E[S_n]| \geq t\right) \leq 2e^{-\frac{2t^2}{\sum_{i=1}^n (b_i - a_i)^2}} \tag{4}$$

*where $S_n = X_1 + \ldots + X_n$.*

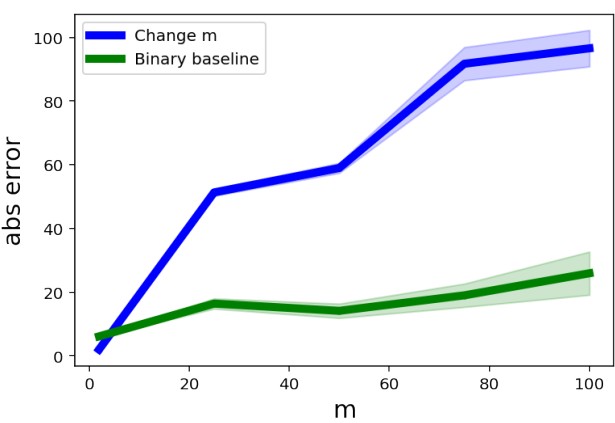

Figure 3: Evaluation of Gemini 1.5 on the MFC task. See Section D.

## D   Empirical Results on the MFC Task

In Section 6.2 we provided results for running Gemini 1.5 on the QC task. Here we provide results for the MFC task. Experimental setting is similar to QC, with the samples drawn in the same way.

The prompt has the following format: "You are given an array of length 5. Find the count of the most frequent element in the array. The array is [1,1,2,2,3]. How many times does the most frequent word in the array appear? Respond in just one number. No additional text.". Another difference from QC was that the sequence length was set such that the expected MFC would be 40, for the given vocabulary size $m$. This was done via simulation (since no closed form expression is available for expected MFC). For the binary baseline in this case we cannot choose an MFC task, because in the binary case the MFC will be larger than our desired expected count (i.e., 40). Instead, we take the binary baseline to be the *minimum* frequency count, with expected minimum equal to 40. Results are provided in Figure 3, and show that results deteriorate with vocabulary size (also with respect to the binary baseline) again suggesting that vocabulary size affects the complexity of this task, as our theoretical results suggest.

