# OpenReview forum: "When Can Transformers Count to n?"
_ICLR.cc/2025/Conference — Submitted to ICLR 2025_

### Official Review · Reviewer_97Ej · 2024-10-28

**Soundness:** 3
**Presentation:** 2
**Contribution:** 3
**Rating:** 6
**Confidence:** 3

**Summary:**

This paper studies transformers' counting capability, focusing on two counting tasks, i.e., QC and MFE.
For analysis, the paper considers transformers' expressive power. The paper shows two possible
solutions for QC, i.e., Histogram and CountAttend, and highlights the relations between the embedding dimension, the model size, the dictionary size, and the sequence length in the two solutions, respectively. For MFE, the paper further
proves bounds on the size of the embedding compared to the size of the dictionary. Experiments evaluate the dependence between the transformer model size and its ability to perform counting tasks both models trained from scratch and a pretrain LLM.

**Strengths:**

1. The paper analyzes the capability of transformers by considering simple yet representative counting problems, which is an interesting perspective.
2. In counting tasks, the paper highlights the relations between the embedding dimension, the model size, the dictionary size, and the sequence length theoretically. These results present an architectural limitation of the transformer and provide insights on how to overcome the issues.

**Weaknesses:**

1. The theoretical analysis is restricted to the expressive power of the transformer. It is unclear whether the proposed solutions for the counting tasks are learnable.
2. While the theoretical analysis shows two possible solutions for QC, the experiments do not demonstrate the mechanisms of the transformer to perform the task.

**Questions:**

1. Do the constructions require infinite precision or only log precision?
2. For QC, which are the mechanisms of the transformer to perform the task in practice?
3. The lower bounds in Lemma 4.4 and Theorem 5.1 are proved with constant-depth models.
Do the lower bounds hold for deep models (up to poly(n) depth)? If not, do the lower bounds hold for constant-depth models with CoT?

---

> ### Author Response · Authors · 2024-11-17
>
> We thank the reviewer for the positive feedback.
>
> Weaknesses:
>
> 1) Our work is indeed focused on the expressive power of transformers, which is very common in the learning theory community, as there are many papers on expressiveness of neural networks. Studying optimization theoretically is out of scope for our paper, although experimentally we show our results for both transformers trained from scratch and pre-trained models (e.g. Gemini 1.5). This indicates that the issues discovered in our paper persists also for optimization.
>
> 2) We agree that it would be interesting to understand how transformers implement counting tasks internally. However, our focus is on when transformers are expected to fail in this task, and our empirical results agree with our theoretical predictions. In these failure cases, the transformer does not carry out the counting task, and therefore mechanistic interpretability becomes less relevant.
>
> Questions:
>
> 1) The constructions require log precision. Namely to count until n, having O(log(n)) bits suffices. Note that the lower bound in Theorem 5.1 specifically includes the number of bits, namely having n bits can break this lower bound. This is not surprising as having very large precision can replace the requirement of having a large embedding dimension by encoding the different coordinates into a single large-precision number.
>
> 2) As noted above, our focus is on understanding when transformers are expected to fail, so we are less focused on solutions that are implemented in success cases. But we agree of course that this is an interesting question for future study.
>
> 3) The lower bound in Lemma 4.4 holds up to depth log(n), although with a smaller width. Showing this limitation requires counting the number of possible linear regions which was done in previous works that study piecewise linear activations. The lower bound in
> Theorem 5.1 is only for depth 1. It uses a communication complexity argument, and it is currently an open problem to extend this lower bound for more than 1-layer. Both of these lower bounds do not hold for CoT prompting, as we did not study this setting. It was shown that unbounded CoT prompting is a universal approximator (Auto-Regressive Next-Token Predictors are Universal Learners, 2023), so extending our lower bounds to this setting requires some bound on the allowed CoT prompting. This will be an interesting direction for future work.

---

> > ### Comment · Reviewer_97Ej · 2024-11-23
> >
> > Thank you for your response and my concerns have been adequately addressed. I will maintain my score.

---

### Official Review · Reviewer_9Adr · 2024-11-03

**Soundness:** 3
**Presentation:** 3
**Contribution:** 2
**Rating:** 5
**Confidence:** 5

**Summary:**

Paper investigates the ability of Transformer models to perform simple counting tasks, specifically focusing on counting the occurrences of tokens in a sequence. It explores whether Transformers can count effectively and, if so, under what conditions.

**Strengths:**

This paper is well-written, presenting a complex topic in a clear and structured manner. the use of well-defined examples, such as the "Query Counting" task and the "Most Frequent Element" problem, aids in illustrating the limitations and possibilities of Transformer architectures in a concrete way.

**Weaknesses:**

Overall, many of the assumptions in this paper are overly simplistic and do not align well with real-world Transformer design or training outcomes. Based on my experience and prior research on Transformer expressiveness, the findings presented here—particularly those concerning cases where d>m have been hinted at in earlier works and are not surprising, given the simplified single-layer Transformer model used in this study. However, real-world Transformer training is far more complex and has been shown to perform poorly on counting tasks, regardless of dimensionality.
Moreover, the experimental design in this paper is weak, with insufficient results to substantiate the authors' claims. The lack of detailed experimental methodology and the vagueness of the reported results undermine the argument. While I agree with the proposed "ideal case" of a histogram approach to counting, this notion is too trivial to warrant significant mention. Actual Transformer training involves many additional considerations, and practical optimization rarely converges to such an ideal case, as evidenced by many experiments I have done and those of other researchers.
I suggest the authors conduct a deeper exploration into the challenges of counting with Transformers. As it stands, the current version provides limited insights into the expressiveness of Transformers, on top of the existing papers.

Detailed problem:
1. Counting Problem in Transformers: Previous research has explored the counting problem in Transformers, but this work lacks a comprehensive review of such studies. A notable contribution to this area is the paper "Language Models Need Inductive Biases to Count Inductively", beyond transformer they also study linear attention such as Mamba and RWKV. Additionally, counting on out-of-distribution (OOD) data often requires incremental dictionary size (training to count to n but testing on count to n+k). Several studies, therefore, focus on the "parity" problem in Transformers, where models count the even or odd occurrences of a token, representing an early form of counting research. For instance, "Overcoming a Theoretical Limitation of Self-Attention" proposes a similar approach to counting as your paper, which is not acknowledged here. A recent paper, "Counting Ability of Large Language Models and Impact of Tokenization," provides an extensive overview of counting in Transformers, which could guide a more thorough literature review on counting with Transformer. There are lots of Parity-related theoretical work and experimental work with Transformers, which study counting, they all need to be discussed.

2. Expressiveness Limitations in Transformers: Prior theoretical and empirical findings suggest that Transformers lack the expressive capacity to act as counter machines. For example, DeepMind's work "Neural Networks and the Chomsky Hierarchy" shows that Transformers struggle even with parity tasks, which are simpler than general counting. Moreover, "Language Models Need Inductive Biases to Count Inductively" and "Overcoming a Theoretical Limitation of Self-Attention" argue that complete Transformer architectures cannot perform counting tasks without specific inductive biases. Common positional encodings (like sinusoidal or absolute encoding) fail in this regard. My own experiments corroborate this—Transformers do not converge to a counting model without cetain biases. This limitation is examined thoroughly in "Overcoming a Theoretical Limitation of Self-Attention," which offers a similar solution to the one proposed in your paper.

3. Limited Theoretical Scope: The theoretical analysis in this paper overlooks several critical components in modern Transformer design that impact expressiveness, such as layer normalization, positional encoding, floating-point precision, residual connections, and advanced activation functions. Many recent studies cover these in greater depth. By simplifying the Transformer to a single attention module, this paper overlooks architectural nuances, which limits the validity of its conclusions. I recommend engaging with recent theoretical work on Transformer expressiveness in light of these elements (matter of fact, many research show that some of the modules here greatly change the theoretical limits of Transformer). The simplified version of your work is similar to MLP, where each node simply connect to every other previous node, and therefore results are trivial.

4. Precision Considerations: Precision is not adequately addressed here. While the paper mentions the need for
d>m it omits the scenario where infinite precision enables Turing completeness with finite d. This has been shown in works like "Attention is Turing Complete" and recent chain-of-thought (CoT) research by Tengyu Ma and others. With higher precision, more information can be compressed within floating-point numbers, contrary to this paper's assumption of clear-cut feature vectors.

5.  The counting discussed in this paper is non-inductive, while most modern large language models (LLMs) rely on inductive counting, requiring
O(N) depth complexity. Both "Language Models Need Inductive Biases to Count Inductively" and "Counting Ability of Large Language Models and Impact of Tokenization" discuss this in detail. This is critical for two reasons: (a) natural language counting often requires sequential comprehension, necessitating a step-by-step approach; and (b) in comparison, recurrent models like RNNs handle counting tasks more naturally than Transformers (almost easily achieve 100\% OOD accuracy according to above papers), highlighting the role of recurrence in inductive counting. Recurrence signals a model’s inductive capacity for counting, suggesting that neural networks generally adopt an inductive counting approach without explicit biases.

6.Prior research shows that positional embedding is crucial for OOD counting. This paper assumes that any positional encoding will equip the model with positional information for counting. However, specific positional encoding designs can distort theoretical assumptions. For example, transformations of values like 1 and 0.5 through positional encoding (either additive in Cosine or geometrical shifting as in ROPE) can cause information loss and lead to unexpected behaviors. These aspects, covered in previous studies, are not discussed here.

6. Experiment lacking specifications. Your experiment does not say how in-distribution training and out of distribution testing is done. Within DIstritbuion, the counting behaves different than OOD, as shown by above mentioned papers.


7. The experimental section lacks clarity regarding in-distribution (ID) training and OOD testing. Counting behaves differently in ID versus OOD contexts, as shown in the referenced papers. Providing explicit details on the data split methodology would strengthen the validity of the results.

8. LLM-Based Counting: Counting in LLMs presents unique challenges. Not only does it involve inductive methods, but factors like tokenization and chain-of-thought (CoT) prompting also affect the theoretical counting capacity of the Transformer architecture. Studies by Tengyu Ma and others in "Counting Ability of Large Language Models and Impact of Tokenization" delve into these factors. The absence of these considerations in your experiments reduces the credibility of the findings. CoT prompting can unexpectedly enhance the expressiveness of Transformers by mimicking recurrent behaviors similar to RNNs. Although the paper discusses LLMs, it does not mention use of CoT and its effect, which is worth addressing given its relevance to theoretical expressiveness.

9 Gap Between Theory and Practice: The theoretical proofs claim the existence of a Transformer capable of counting, albeit simplified. However, existence does not guarantee practical trainability. As previously mentioned, Transformers rarely converge to this type of behavior in practice without specific biases. Floating-point representations of features can also complicate interpretation, indicating a substantial gap between theoretical assertions and empirical outcomes. More experimental evidence is needed to substantiate the theoretical claims made in this paper.

**Questions:**

To clarify the unique contributions of your work compared to "Overcoming a Theoretical Limitation of Self-Attention," it would be beneficial to highlight additional perspectives or distinct aspects that differentiate your approach.  As they propose nearly identical Transformer design in counting but with more in-depth experimental analysis.

---

> ### Author Response · Authors · 2024-11-17
>
> We thank the reviewer for the thorough review.
>
> We believe that the reviewer missed the main point of the paper. Namely, that transformers are limited when performing simple counting tasks for large vocabulary sizes.
> Our results show that simple solutions fail when the dictionary size is large (Theorem 4.2), and that any solution to the MFE tasks requires the embedding dimension to depend on the dictionary size (Theorem 5.1). The histogram solution that the reviewer mentions is not the main result of our paper. It just shows that there exists a simple solution when the dictionary size is small.
>
> In particular, papers about the parity problem (e.g. Overcoming a Theoretical Limitation of Self-Attention) only deal with dictionaries of size 2, while our paper focuses on problems where the dictionary size is large. Specifically, our theoretical results predict that counting would become harder as vocabulary size grows. And, our empirical results are in very good agreement with this, and also agree with the phase transition predicted by theory.
>
> To answer the specific issues:
>
> 1) The histogram solution that the reviewer mentions is only applicable  in unrealistic situations when the embedding dimension is larger than the dictionary size. In realistic settings, we show the limitations of transformers to solve such tasks. Also note that the parity problem is inherently different from counting when the dictionary size is large, thus the works on parity and other bit-related problems are not comparable to our work. With that said, we will add “Overcoming a Theoretical Limitation of Self-Attention” to the related works section.
>
> 2) We indeed agree that transformers lack the expressive ability to count, and as far as we know, ours is the first work that shows this limitation theoretically for large vocabulary sizes (compared to vocabulary size of 2 in bit-related problems). One important thing to notice is that transformers DO NOT have expressivity problems when it comes to counting a small vocabulary. This is precisely what our histogram solution shows. Specifically, counting bits is straightforward (e.g. if the input is 0,1 then summing it will produce the count. There is still of course the need to translate the count to the set of right tokens, but in our case the expressivity issue arises even without this requirement). The reviewer should note that counting bits is very different from the parity task, where there is the added challenge of calculating the parity of the count.
>
> 3) Our theoretical analysis does in fact make use of positional embeddings and precision (see Theorem 5.1 which has a limitation on the required bit-precision). We also mention layer normalization in lines 159-160 and that it does not affect our theoretical results. We also mention residual connections several times and make use of them (see lines 198, 231-232). We indeed focus on the ReLU activation which is done in many previous works. Also, in the experimental framework we use all the above components. We do not understand this criticism as our work covers all of them, both theoretically and empirically.
>
> 4) We specifically make use of finite-bit precision in Section 5, and Theorem 5.1 includes the term “p” which is the precision of the model. Theorem 4.1 and the histogram solution can work with O(1) bit-precision, independent of the context length or dictionary size.
>
> 5) The reviewer writes: “The counting discussed in this paper is non-inductive, while most modern large language models (LLMs) rely on inductive counting, requiring O(N) depth complexity.” We ask that the reviewer define exactly what they view counts as “inductive counting” and in what sense “most modern large language models (LLMs) rely on inductive counting”. Our paper provides a clear definition of the counting problem we address, and we see no clear argument for why it is not an interesting counting problem. Specifically, we do not agree that counting to N requires O(N) depth, because our histogram construction shows that one can always solve the query-count task for vocabulary size m, as long as the model dimension d satisfies d>2m, with one transformer layer. This refutes the claim by the reviewer that O(N) depth is needed.
> The reviewer states that RNNs have a natural mechanism for counting (via their state). This is true, but we don’t see how it is relevant when one studies how transformers (which currently underlie all successful LLMs) count.
> We also note that the reviewer mentions two papers in this comment where one (“Language Models Need Inductive Biases to Count Inductively”) is under review at this ICLR and the other (“Counting Ability of Large Language Models and Impact of Tokenization”) was put on arxiv AFTER the ICLR deadline. Our paper is in any case very different from these, because our focus is on the interplay between vocabulary size and model dimension. However, in any case, these would not count as prior work.

---

> > ### Author Response · Authors · 2024-11-17
> >
> > 6) Our work is constructive in nature, and we construct a specific positional embedding that solves the counting tasks. We don’t say anywhere in our paper that any positional embedding can work, because that is obviously not true.
> >
> > 7) We agree that OOD experiments may differ, but since this is more of a theoretical work in nature, we haven’t done this experiment and studied standard supervised learning tasks where the train and test come from the same distribution. There are many different experiments that can be done. However we think it shouldn’t be expected from a theoretical work to conduct all of them. Our experiments clearly validate our main theoretical prediction, which is that counting becomes harder for transformers as the vocabulary size grows.
> >
> > 8) We do experiments with pre-trained LLMs (e.g. Gemini 1.5) as seen in the experiments section, and include all the features that the reviewer mentioned including tokenization. We indeed do not study chain-of-thought prompting, which is well out of scope for this work.
> >
> > 9) We think that the reviewer missed the main point of our paper. Our theoretical results imply that transformers are limited in solving counting tasks with large vocabulary sizes, which is verified by our experimental section. We are not sure what is the gap that the reviewer mentions.

---

> ### Comment · Reviewer_9Adr · 2024-11-18
> **reply to authors**
>
> Thanks for your comment, my concern are all addressed and I have changed my scores, the only thing I am still worried about Is experiments, as mentioned in my review, as theoretical side now makes sense to me.

---

> > ### Author Response · Authors · 2024-11-18
> >
> > Thank you for your prompt response, and for raising your score. Regarding the experiments, we understand that OOD is a further challenge for counting in transformers. However, the setting we study already highlights challenges for within distribution learning. It seems like OOD evaluation will indeed make the task even more challenging, but possibly due to other reasons.
> > Thus, we thought it is better in terms of presentation simplicity to not introduce additional variations on the evaluation scheme. We are happy to add details if requested.

---

### Official Review · Reviewer_yQxr · 2024-11-04

**Soundness:** 2
**Presentation:** 3
**Contribution:** 2
**Rating:** 6
**Confidence:** 5

**Summary:**

This work investigates the ability of LLMs in counting tasks. This paper focuses on two basic tasks, QC and MFE, and demonstrates that the transformers can solve these tasks if the models' size is large enough, but face limitations when the dimension is small. Empirical and theoretical results highlight the importance of understanding these limitations to improve transformer architectures.

**Strengths:**

1. This work focuses on the counting task for the language models. The authors provide both theoretical and empirical results to demonstrate the limitations of LLMs when the dimension is small.
2. This paper is well-presented and easy for the readers to follow.

**Weaknesses:**

1. **Lack of Generality:** While the paper focuses on the counting task, its impact on real-world applications is unclear. The conclusions are specific to counting and may not generalize well to broader contexts.
2. The study primarily analyzes one-layer transformers, leaving the capabilities of multi-layer transformers unexplored. Further **theoretical** investigation is needed to understand how additional layers might influence performance on counting tasks.

**Questions:**

1. How can the conclusions of this paper generalize to real-world tasks, such as math, code, and so on?
2. Do the limitations observed with small dimensions still apply to multi-layer transformers?

---

> ### Author Response · Authors · 2024-11-17
>
> We thank the reviewer for the thorough review. We address the comments below.
>
> Weaknesses:
>
> 1) Our work focuses on simple counting tasks, and shows that even for such simple tasks there are inherent limitations of the transformers architecture for large vocabulary sizes. We believe this is an important stepping stone in the study of transformers. For comparison, there are many works on neural networks studying specific tasks that greatly helped the community in understanding the strengths and limitations of such architectures. Some examples include depth separation for specific target functions [Telgarsky, 2016], [Eldan & Shamir, 2017], learning parity under different distributions [Malach & Daniely 2020], learning single neurons using kernels [Yehudai & Shamir 2019], [Kamath et al. 2020] and many more. Additionally, counting is a core capability that underlies more elaborate tasks (e.g., summarization, data analysis). Thus, we believe it is very important to understand when transformers can achieve this task.
>
> 2) Indeed our theoretical results are limited to 1-layer of attention, although we do allow an arbitrary sized MLP (for example in Theorem 5.1). Our result that 1-layer transformers cannot count (e.g., for MFE) when vocabulary size grows is novel and surprising, because it shows the special role of vocabulary size in this task. We use a communication complexity argument, and it is an open problem on how to extend it to multi-layer transformers. We are currently not aware of any theoretical work showing expressive limitation of transformers for more than 2-layers, and showing such a result would have great impact.
>
> Questions:
>
> 1) We believe that if transformers are limited in solving such simple counting tasks, then solving even more complex tasks would be even more limited. Our paper focuses on such tasks since these can be analyzed theoretically, and we are the first work, as far as we know, to show theoretically the limitations for solving such simple counting tasks when the vocabulary size is big. It would be interesting to formalize and theoretically analyze other tasks related to code and math, although we suspect it would be hard.
>
> 2) Empirically, the limitations are also manifested in multi-layer transformers, and also large pretrained models (e.g. Gemini 1.5), as depicted in the experiments section. Theoretically, it would be very interesting to show expressiveness limitations for multi-layer transformers. This issue is also discussed in Representational strengths and limitations of transformers (Sanford et al. 2023).

---

> ### Comment · Reviewer_yQxr · 2024-11-28
> **Thanks for your response**
>
> Thanks for your response. My concerns have been addressed, I will increase my score.

---

### Official Review · Reviewer_UpN2 · 2024-11-04

**Soundness:** 4
**Presentation:** 3
**Contribution:** 3
**Rating:** 6
**Confidence:** 3

**Summary:**

This paper presents a theoretical study on how a Transformer can learn the solution of counting. They consider two typical solutions. The first one is a histogram solution that keeps a histogram of the number of different types of tokens, and this requires a dimension linearly scaling with the vocabulary size. The second solution is by first calculating the inverse of the number of times the query tokens appears and then use the MLP to calculate the inverse function. Intriguingly, they show if the feedforward layer is of depth 1, the required width to represent the inverse function scales linearly with the context length.They empirically verified their theory on pretrained LLMs.

**Strengths:**

1. The presented theory is very clear. The authors explain their theoretical contribution with very intuitive argument.

2. The theoretical argument that vocabulary size and context length jointly blocks the learning of counting is well supported by empirical experiments.

**Weaknesses:**

1. The work is mostly constructive so it remains unclear whether Transformers will converge to either of the solution. A mechanistic investigation as mentioned in the conclusion will be a great supplement for the paper.

2. The width bottleneck in the second construction seems to hold only for 1-layer MLP.

3. Technically, the argument that position encoding is necessary only holds for encoder-based model or causal model with 1-layer, a point that should be made clear in the paper.

**Questions:**

1. If LayerNorm is used in the architecture, could this generate more parameter efficient architecture?

2. Is there a way to investigate what solution Transformers really converge to in training other than probing? For example, will the two construction differs meaningfully on some out of distribution test?

---

> ### Author Response · Authors · 2024-11-17
>
> We thank the reviewer for the positive feedback. We address the comments below.
>
> Weaknesses:
>
> 1) Indeed our paper focuses on the expressive power of transformers from a theoretical point of view, as was done in several previous works. Clearly it is possible to also pursue a mechanistic interpretability study of counting, and our results indeed suggest what these mechanisms might be. However, we believe this would make sense as a separate contribution, and our results here are of sufficient interest, as they point to core limitations of transformers for this important task.
>
> 2) The width bottleneck in the CountAttend construction is indeed presented only for 1-layer MLP. It can be readily extended to deeper MLPs at the cost of having a smaller width constraint, and we will add a remark about this in the final version. Namely, the bottleneck means that the construction requires $n$ linear pieces, which can be done using an $L$-layer MLP with width $m$ as long as $2^L\cdot m \geq \Omega(n)$ (this is a standard result about counting the number of linear pieces for a ReLU MLP. See e.g., “On the Number of Linear Regions of Deep Neural Networks, 2014”). This means that if $L=1$ (as we have in the result) the width needs to be $\Omega(n)$, but it can also be done with depth $\log(n)$ and width $O(1)$. We chose to focus on small and fixed depth, as that is the common implementation in transformers.
>
> 3) This is a good point, we will make it clear in the final version.
>
> Questions:
>
> 1) Thanks for the great question. In the experimental section we do use a standard architecture with normalization layers, and the effect is evident there, so LayerNorm does not seem to help in solving the task. In the theory part, the MFE lower bound still holds with LayerNorm because the lower bound does not put any restriction on the MLP size, and an MLP can be used to implement LayerNorm (e.g. by adding neurons that compute the square of their input, which can be done with $\log(eps^{-1})$ layers, using the construction from Telgarsky 2016).
>
> 2) This sounds like a good suggestion for additional experiments to gain insight into internal implementation. Namely, training on one distribution of tokens (e.g. i.i.d), and testing on a different one (e.g. when different tokens have different probabilities). We chose to focus on the simplest empirical settings, as they already demonstrate our key result (i.e., that increasing vocabulary size makes counting harder for transformers). We will consider adding such an experiment in the final version.

---

> > ### Comment · Reviewer_UpN2 · 2024-11-24
> >
> > I have read the response and I will keep my positive rating.

---

### Official Review · Reviewer_i79M · 2024-11-09

**Soundness:** 2
**Presentation:** 3
**Contribution:** 2
**Rating:** 5
**Confidence:** 4

**Summary:**

This paper studies an important question on the counting ability of Transformers. An interesting construction is proposed to address the query counting problem using Transformer architecture.

**Strengths:**

This paper studies an important question on the counting ability of Transformers. An interesting construction is proposed to address the query counting problem using Transformer architecture.

**Weaknesses:**

Although some theoretical discussion are provided for the proposed construction, the construction itself is only a toy model and may be too simple to reflect the ability of realistic Transformers. Also, the fact that this particular construction cannot achieve certain tasks does not indicate that there does not exist a construction that can. Plus, there are too many loose ends in the proofs (see Questions below).

**Questions:**

1. In your examples, you are counting the number of appearance of certain letters instead of tokens. Therefore, tokenization plays an important part in this task. However, tokenization is not discussed in the paper.
2. The proposed architecture only considers one single layer with one head without normalization layers. This is not the standard Transformer architecture with MLP and skip connections.
3. Lines 181-183 argue that replication of the input results in the same output. Is there any theoretical proof? Is this true in practice?
4. Please elaborate on why the assumption in Eq(1) holds in reality.
5. In the entire paper, including the experiment section, the training of the model and the training data are not discussed. Do pre-training, training data, and fine-tuning affect the ability of Transformers? What if a dataset is constructed with (sequence, count) pairs?

---

> ### Author Response · Authors · 2024-11-17
>
> We thank the reviewer for the thorough review. We address the comments below.
>
> We first emphasize that the focus of our paper is the interplay between vocabulary size and embedding dimensionality in counting tasks. Our theoretical results suggest that counting will be harder for transformers as the vocabulary size grows, and this is supported by our empirical results. Second, our lower bounds are general and apply to any construction, not specifically the ones we provide. For example, in Theorem 5.1 we prove that **any** one-layer transformer cannot solve the MFE task unless the embedding dimension is large enough.
>
> Regarding the questions:
>
> 1) Our theoretical constructions and lower bounds are designed to count tokens rather than words or letters. This applies to both the QC and MFE tasks. To simplify the presentation of the task we use a different letter for each possible token. In our experiments, for the transformers trained from scratch, we used a similar setting as in the theoretical section where each letter is presented as a token. For the pre-trained transformer (i.e. Gemini 1.5) we used the standard tokenization used by the model.
>
> 2) Throughout the paper, we use an MLP for all of our constructions. This is also mentioned in the theorem statements. Our constructions and lower bounds can be readily extended to include skip connections, we haven’t used them as they are unnecessary for the upper bounds and don’t affect the lower bounds. We will add them if the reviewer thinks it will strengthen the results. Regarding the number of layers, indeed our lower bound in Theorem 5.1 is restricted to 1-layer of attention (although with an MLP that has any number of layers). We use a communication complexity argument for the proof, and it is an open problem on how to extend this proof method for multi-layer transformers, which is also discussed in previous works on the theory of transformers (see e.g. Representational strengths and limitations of transformers, 2023)
>
> 3) This is a simple and straightforward fact that we mention informally and was also made in a recent work (Barbero et al. 2024). We can add more formal proof to it. It does happen in practice as it is a limitation of the architecture itself. However, note that this limitation only happens when the model has no positional embedding. Adding even the simplest positional embedding (e.g., by marking the last token) solves this issue.
>
> 4) This assumption is only part of the expressiveness result presented in Theorem 4.1 when the input dimension is larger than the dictionary size. This often **does not** hold in practice, because the dictionary size is several times larger than the embedding dimension. This is also not a main assumption in our paper and is only relevant for this specific result. For this reason, in Section 4.3 we describe how in a more realistic setting the histogram solution fails and a more complex solution is needed (i.e. the CountAttend).
>
> 5) For the experiments, all the training details are presented in Section 6, including data generation, and implementation details. In Section 6.2 we specifically discuss pretrained transformers, which still struggle with counting tasks when the dictionary size is large. In Section 6.1 we discuss transformers trained from scratch on this specific task, which is stronger than just fine-tuning, and the problem with counting over large vocabulary persists. Indeed, in this training, we provide supervision of (sequence, count) pairs (corresponding to x1,...,xn and y described in the text, in Tasks section of Section 6.1). Our results in all these settings demonstrate a clear effect of vocabulary size on the ability to count, as predicted by our theoretical analysis.
>
> The reviewer mentioned loose ends in the proofs, we would be very happy if the reviewer could point to specific such points so that we could address these in our response.

---

### Author Response · Authors · 2024-11-17
**Response to all the reviewers**

We thank all reviewers for their comments. We would like to emphasize a general point about our submission: our main focus is the interplay between vocabulary size and embedding dimension for counting tasks. Namely, we ask how difficult it is to count when the number of unique elements you count (i.e., the vocabulary size) changes. Note this is very different from works that looks at counting the number of bits in a sequence, or the parity problem. Specifically, if the inputs are just {0,1}, then counting the number of bits can simply be done by summing the inputs, with a transformer with one layer.

It was not clear to us, when we began working on this, whether vocabulary size should have an effect on the difficulty of the task at all. However, both our theoretical and empirical results show that it has a pronounced effect. This is a new result that we believe will be of considerable interest to the community.

---

### Meta-Review · Area_Chair_wjzA · 2024-12-21

**Metareview:**

As the title suggests, this paper investigates the counting capabilities of Transformer models, focusing on simple counting tasks such as Query Counting (QC) and Most Frequent Element (MFE). The authors claim that the ability of transformers to perform these tasks is significantly influenced by the vocabulary size and the embedding dimension. They provide theoretical results showing that the embedding dimension must scale with the vocabulary size for effective counting. Empirical results support these theoretical findings, demonstrating a phase transition in performance as predicted.

Strengths:
- **Novel Theoretical Insights**: The paper provides new theoretical insights into the limitations of Transformers for counting tasks, particularly highlighting the impact of vocabulary size.
- **Empirical Validation**: The theoretical claims are supported by some empirical results, which show a clear phase transition in performance.
- **Clarity and Presentation**: The paper is well-written and presents complex theoretical concepts in a clear and structured manner.

Weaknesses:
- **Simplistic Assumptions**: The theoretical analysis is based on overly simplistic assumptions (eg one-layer transformer) that may not hold in real-world scenarios.
- **Limited Experimental Scope**: The experiments are not comprehensive enough to fully support the theoretical claims. Specifically, the lack of OOD evaluation and detailed experimental methodology weakens the empirical evidence.
- **Lack of Mechanistic Insights**: The paper does not explore the mechanisms by which Transformers perform such counting tasks, which could provide deeper insights into the model's capabilities and limitations, and leaves many questions open.
- **Concerns about Validity of Experimental Setting**:

Most important reason to reject: overly simplistic model with strong assumptions and limited generality, and therefore likely to have modest impact in the community.

Overall, this seems to be a borderline paper, with interesting somewhat novel results and insights that I believe might not raise to the level of acceptance to ICLR.

**Additional Comments On Reviewer Discussion:**

The main issues raised during the rebuttal period focused on the simplicity of the model and the underlying assumptions. The authors provided thoughtful responses to most of these points, but many of the reviewers remain unconvinced. In particular,

- **Reviewer i79M**: Highlighted the simplistic nature of the theoretical model and the lack of discussion on tokenization and training data. The reviewer did not respond to the rebuttal, and their concerns remained largely unaddressed.
- **Reviewer 9Adr**: Initially raised concerns about the experimental design, the relevance of the theoretical results, and some related work. Although the reviewer acknowledged the detailed rebuttal and increased their score, they maintained that the experimental evidence was insufficient.
- **Reviewer UpN2**: Appreciated the theoretical contributions but noted the need for mechanistic investigations and more comprehensive experiments. The reviewer maintained a positive score after the rebuttal.
- **Reviewer yQxr**: Raised concerns about the generality of the results and the lack of exploration of multi-layer Transformers. The reviewer increased their score after the rebuttal, indicating that their concerns were addressed.
- **Reviewer 97Ej**: Questioned the learnability of the proposed solutions and the applicability of the theoretical results to deep models. The reviewer maintained their score after the rebuttal.

Note: concerns about related work for concurrent or posterior work were heavily downweighed in my assessment, and were used only as reference for the type of broader evaluation that could have been performed in this paper. As stated above, perceived novelty was not the main reason for rejection.

---

### Decision · Program_Chairs · 2025-01-22

Reject